**Transverse jointing in foreland fold-and-thrust belts: a remote sensing**
**analysis in the eastern Pyrenees**
Stefano Tavani[1], Pablo Granado[2], Amerigo Corradetti[3], Thomas Seers[3], Josep Maria Casas[2],
Josep Anton Muñoz[2]
1 - DISTAR, Università degli Studi di Napoli "Federico II", Via Cupa Nuova Cintia 21,
80126, Naples, Italy
2 - Institut de Recerca Geomodels, Departament de Dinàmica de la Terra i de l'Oceà,
Universitat de Barcelona, C/ Martí i Franques s/n, 08028, Barcelona, Spain
3 - Department of Petroleum Engineering, Texas A&M University at Qatar. Education City,
PO Box 23874, Doha, Qatar
**Abstract**
Joint systems in the eastern portion of the Ebro Basin of the eastern Pyrenees enjoy near
continuous exposure from the frontal portion of the belt up to the external portion of its
associated foredeep. Utilizing orthophoto mosaics of these world class exposures, we have
manually digitized over 30000 joints within a 16x50 km study area. The mapped traces
exhibit orientations that are dominantly perpendicular to the trend of the belt (transverse) and,
subordinately, parallel to the belt (longitudinal). In particular, joints systematically orient
perpendicular to the trend of the belt both in the frontal folds and in the inner and central
portion of the foredeep basin. Longitudinal joints occur rarely, with a disordered spatial
distribution, exhibiting null difference in abundance between the belt and the foredeep. Joint
orientations in the external portion of the foredeep become less clustered, with adjacent areas
dominated by either transverse or oblique joints. Our data indicates that joints in the studied
area formed in the foredeep in response to a foredeep-parallel stretching, which becomes
progressively less intense within the external portion of the foredeep. There, the minimum
stress direction becomes more variable, evidencing poor contribution of the forebulge-
perpendicular stretching on stress organization.

**Introduction**
Fractures can be effective pathways for fluid flow (e.g. Laubach et al., 2019), thus
impacting production of hydrocarbons (Barr et al., 2007; Engelder et al., 2009; Questiaux et
al., 2010) and geothermal water (Haffen et al., 2013; Vidal et al., 2017), the pathways and
fates of contaminants released from deep geological radioactive waste repositories
(Berkowitz et al., 1988; Iding and Ringrose, 2010) and the sustainable management of
groundwater (Masciopinto and Palmiotta, 2013). Associated with crustal tension, joints are
ubiquitous open-mode fractures occurring in a range of tectonic settings, including collisional
belts. In collisional settings, layer bending and stretching during the growth of thrust-related
anticlines has conventionally been invoked as the principal causative process for the
development of joints oriented approximately parallel (e.g. Ramsay, 1967; Murray, 1968) and
perpendicular (e.g. Dietrich, 1989; Lemiszki et al., 1994) to the trend of the belt and of the
thrust related anticlines. However, the frequently observed obliquity between joints (and
other meso-scale structures) and the trend of the hosting anticlines (e.g. Tavani et al., 2019;
Beaudoin et al., 2020), along with the documented occurrence of joints in exposed forelands
(e.g. Dunne and North, 1990; Zhao and Jacobi, 1997; Billi and Salvini, 2003; Whitaker and
Engelder, 2006), has more recently led to the conclusion that, in many cases, joints and other
kinds of extensional fractures exposed in thrust and fold belts have developed prior to folding
and thrusting, within the foreland region (e.g. Doglioni, 1995; Zhao and Jacobi, 1997;
Tavarnelli and Peacock, 1999; Lash and Engelder, 2007; Branellec et al., 2015; Basa et al.,
2019; Giuffrida et al., 2019; Martinelli et al., 2019; Carrillo et al., 2020), where joints are
layer-perpendicular and commonly oriented parallel (longitudinal) and perpendicular
(transverse) to the belt-foredeep-forebulge trend (Tavani et al., 2015).
A partially unresolved question in foreland deformation relates to the development of
transverse joints, which requires a tensile minimum stress oriented parallel to the foredeep.
Even in arched systems, the forebulge, the foredeep, and the belt tend to be nearly parallel to
each other locally (e.g. the Hellenic arc, the Apennines-Calabrian arc, the Betic-Rif arc). The
shortening direction in the inner portion of the foredeep (subjected to layer parallel
shortening) and the stretching direction in the forebulge (where bulge-perpendicular
stretching induced by lithospheric outer arc extension operates) are nearly parallel in a belt-
perpendicular transect (Fig. 1). In addition, in the innermost portion of the foredeep, where
layer parallel shortening operates, the $\sigma 2$ is typically vertical and the $\sigma 3$ is positive,
horizontal, and parallel to the trend of the belt (Tavani et al., 2015). This is evidenced by the
occurrence of bedding-perpendicular pressure solution-vein pairs (e.g Railsback and
Andrews, 1995; Evans and Elmore, 2006; Quintà and Tavani, 2012; Weil and Yonkee, 2012)
and/or conjugate strike-slip faults at a high angle to bedding (e.g. Marshak et al., 1982,
Hancock, 1985, Erslev, 2001; Lacombe et al., 2006; Amrouch et al., 2010, Weil and Yonkee,
2012) occurring in foreland areas and in the adjacent fold and thrust belts worldwide,
although in many cases structures associated with this strike-slip regime do not develop
during layer parallel shortening (e.g. Soliva et al., 2013). Given the above, there should be an
area in between the inner portion of the foredeep and the peripheral bulge where $\sigma1$ becomes
vertical and where $\sigma3$ is still horizontal and parallel to the belt (Fig. 1a). This scenario could
explain the development of layer-perpendicular transverse extensional structures, with
transverse extensional faults or veins expected to develop where $\sigma3$ is positive. In this
scenario, transverse joints occurring in this zone of localized tension could only develop as
cross-joints (e.g. Gross, 1993) of the longitudinal set formed in the forebulge. This simplified
model does not explain the documented occurrence of transverse joints in areas where
longitudinal joints do not occur or represent the cross-joint set (e.g. Zhao and Jacobi, 1997;
Quintà and Tavani, 2012). This framework does not admit a simple stress permutation in the
foredeep and requires a negative (tensile) $\sigma3$ connected to a foredeep-parallel stretching
component (Fig. 1b). Lithospheric bending of the foredeep, both along the horizontal plane
(e.g. Doglioni, 1995) and along the vertical plane parallel to the trench (Quintà and Tavani,
2012), has been invoked as a process able to produce foredeep-parallel stretching.
Continuous exposures across the entire foreland region of the eastern Pyrenees allows
investigation of the primary mechanism responsible for transverse joint development
described above. Tens to hundreds of meters long joints affect the sedimentary sequence of
the Ebro foredeep basin (Fig. 2a), and are found tilted within the frontal structures of the
Pyrenean belt (Fig. 2b). These pre-folding joints are exceptionally exposed and mappable
from orthophotos (Fig. 2c-f), from which they can be traced almost continuously from the
external portion of the foredeep until the thrust belt. We have remotely mapped 30059 joints
traces from the aforementioned orthophoto dataset and obtained their azimuthal distributions
across the study area. Subsequently, this extended lineament database has been used to
constrain the causative mechanism behind transverse jointing in the Ebro foredeep basin.

## 1 Geological framework

2       The study area is situated in the eastern Ebro foreland basin within an area connecting

the eastern Pyrenees with the Catalan Coastal Ranges (Fig. 3a). The Pyrenees is an EW-
striking orogenic system that formed as the Iberian and European plates collided from Late
Cretaceous to Miocene times (Roest and Srivastava, 1991; Rosenbaum et al., 2002; Muñoz,
1992, 2002), and constitutes an asymmetric, doubly vergent orogenic wedge above the
northward subduction of the Iberian lithosphere beneath the European plate (Chevrot et al.,
2018). As a result, the Ebro basin formed as a flexural foreland developed on the downgoing
Iberian plate at the southern margin of the chain (Beaumont et al., 2000). In the study area, to
the south of the Ebro Basin (Fig. 3a), the Catalan Coastal Ranges developed as a Paleogene
intraplate left-lateral transpressional system (López-Blanco, 2002; Santanach et al., 2011).
The low-displacement character of thrusts and the absence of an associated foredeep both
evidence the limited importance of this range. A series of NE-SW and NW-SE trending
extensional faults strike parallel and perpendicular to the Catalan Coastal Range respectively,
which are a result of the late Oligocene-Miocene opening of the NW Mediterranean basin
(Vegas, 1992; Sàbat et al., 1995; Granado et al., 2016a).

17       The study area where joint traces have been digitized is delimited to the north by the

frontal Pyrenean thrusts and by the Eocene Bellmunt anticline (Figs. 3a,b). This anticline
comprises the Paleocene to upper Eocene foredeep infill. Immediately to the south of the
anticline (i.e.< 5 km), this multilayer becomes sub-horizontal and thins southward where the
Paleozoic to Mesozoic foredeep floor gently rises up and becomes exposed (Fig. 3a). There,
this pre-orogenic succession is slightly tilted to the north by uplift in the footwall of NW-SE
striking Ḻlate Oligocene to Neogene extensional faults. Further to the southwest, this
succession is affected by the Paleogene contractional structures of the Catalan Coastal
Ranges (Fig. 3b).

26       In the field, joints are constantly bedding-perpendicular, regardless of the bedding dip

(Fig. 2a,b), and they are characterized by the occurrence of either a single set (Fig. 2c) or by a
ladder pattern (Fig. 2d,e). In the latter case, the few E-W striking joints are almost
everywhere perpendicular to the N-S striking set and abut on it (Fig. 2e). This indicates that
E-W striking joints are cross-joints formed perpendicular to, and about synchronously with,
the N-S striking joint set and that N-S joints formed in response to a negative (tensile)
minimum stress (e.g. Bai and Gross, 1999; Bai et al., 2002).

**Data and Methods**

Joints have been digitized within the 16.4x49.2 km area displayed on Figure 3, on 25 and 50
cm/px orthophotos provided by the Spanish Instituto Geográfico Nacional via the PNOA
(Plan Nacional de Ortofotografía Aérea) project (https://pnoa.ign.es/). The open source
geographical information system QGIS 3.4 has been used to manually digitize 30059 joint
traces. A collection of these traces seen on orthophotos is provided in Figure 2c-f. Joints have
been digitized in Bartonian and, subordinately, in Lutetian and Priabonian sedimentary rocks
(Fig. 3c). The NE and SE portions of the study area are highly vegetated (Fig. 3d,e) and only
a few joint traces have been mapped there. Quaternary sediments unaffected by joints crop
out in the central portion of the study area (Fig. 3d).
Digitized traces have lengths ranging from 2 to 100 m, with an average of ~20 m (Fig.
3f). The frequency distribution of trace trend shows that the vast majority of the mapped
joints are approximately N-S striking (Fig. 3f). A second subordinate set corresponds to E-W
striking joints, which in the field occur as cross-joints (Fig. 2d). The frequency distribution of
trace trend is not symmetrical around these orthogonal sets, due to the presence of a third less
abundant set composed of NW-SE oriented joints (Fig. 2e,f). The NW-SE striking joints are
mostly seen in the southern portion of the study area, in exposures where the N-S striking set
is not occurring. As mentioned above, exposures are frequently characterized by a single set
(Fig. 2c,d). At many locations, the dominant set is accompanied by associated cross-joints
(Fig. 2d). Very rarely, two mutually oblique sets occur in the same exposure (Fig. 2f).
In order to evaluate the variability of joint traces, the 16.4x49.2 km study area has
been divided into meshes of equilateral triangular elements with edge lengths of 1025m
(Mesh 1) and 1640m (Mesh 2). At each node, mean value and variance of trace trends has
been computed using a circular moving window with a radius of 1200m (Mesh 1) and 1900m
(Mesh 2). The radius of the circular moving window is set to these values for two reasons: 1)
it is two orders of magnitude longer than the average length of joints; 2) it is large enough to
ensure that most of the nodes have data number >20, as only nodes with data number > 20
have been analyzed. Since trace trends are circular data with an angle ($\alpha$) over a period of
180°, in agreement with Mardia (1975) we used equations 1 to 4 to derive at each node the
circular mean value ($Mv_\pi$), the circular variance ($V_\pi$) and the resultant length ($R_\pi$; $V_\pi = 1-R_\pi$),
the latter spanning from 0 (unclustered distribution) to 1 (perfectly clustered distribution). In
the presence of cross-orthogonal joint sets, it is also useful using a period of $\pi/2$, thus
modifying Mardia's equations and introducing the $Mv_{\pi/2}$ and $R_{\pi/2}$ parameters, which are
computed using Equations 5 to 8.

$$C_\pi = \frac{\sum_{i=1}^n \cos(2\alpha_i)}{n} \ (1) \ ; \ S_\pi = \frac{\sum_{i=1}^n \sin(2\alpha_i)}{n} \ (2) \ ; \ V_\pi = 1 - R_\pi = 1 - \sqrt{C_\pi^2 + S_\pi^2} \ (3) \ ; \ Mv_\pi = \frac{Arc\tan\left(S_\pi/C_\pi\right)}{2} \ (4)$$

$$C_{\pi/2} = \frac{\sum_{i=1}^n \cos(4\alpha_i)}{n} \ (5) \ ; \ S_{\pi/2} = \frac{\sum_{i=1}^n \sin(4\alpha_i)}{n} \ (6) \ ; V_{\pi/2} = 1 - R_{\pi/2} = 1 - \sqrt{C_{\pi/2}^2 + S_{\pi/2}^2} \ (7) \ ; \ Mv_{\pi/2} = \frac{Arc\tan\left(S_{\pi/2}/C_{\pi/2}\right)}{4} \ (8)$$

By using these four parameters together, instead for example of the classical k-means
clustering analysis, it is possible to derive important considerations on the distribution of
polymodal distributions in which two mutually orthogonal sets do occur, as illustrated in
Figure 4. We compare the Mv and the R parameters computed using the $\pi$ and $\pi/2$ periods. In
the first example two parallel traces are analyzed, resulting in $Mv_{\pi/2} = Mv_\pi$, and $R_{\pi/2} = R_\pi = 1$
(i.e. circular variance = 0). When data dispersion is slightly increased (Example 2), $Mv_{\pi/2}$ is
still equal to $Mv_\pi$, whereas $R_{\pi/2}$ decreases faster than $R_\pi$. Further increase of data dispersion
(Example 3), in an asymmetric distribution (i.e. non-orthogonal sets), causes additional
decrease of $R_{\pi/2}$ with respect to $R_\pi$, and slight divergence between $Mv_{\pi/2}$ and $Mv_\pi$. In the
presence of a cross-orthogonal subset, the statistical usefulness of $R_{\pi/2}$ becomes evident, as
illustrated in Example 4. In this case, $R_\pi$ rapidly approaches zero, suggesting high dispersion
(i.e. unrepresentative $Mv_\pi$), whereas $R_{\pi/2}$ is essentially unaffected with respect to Example 2,
indicating low dispersion and a representative $Mv_{\pi/2}$. However, the use of the $\pi/2$ period only
returns results in the 0 to $\pi/2$ range, so that NW-SE trending traces result in a NE-SW
trending mean value ($Mv_{\pi/2}$), as shown in the Example 5. In summary, $Mv_\pi$ is useful to derive
the mean direction, whereas $R_{\pi/2}$ and $R_\pi$ should be used in conjunction to discriminate
between populations in which oblique sets occur ($R_{\pi/2} < R_\pi$) from those in which two
perpendicular sets occur ($R_{\pi/2} > R_\pi$).

**Results**
Figure 5 displays attribute maps generated from Mesh 1 and Mesh 2 using the trace
orientation parameters described above. Both Mesh 1 and Mesh 2 have $R_\pi > 0.5$ across almost
the entire study area, with the only exception being in its NW corner. For $R_{\pi/2}$, in addition to
the NW corner, the central portion of the study area has $R_{\pi/2} < 0.5$. Noteworthy differences
between $R_\pi$ and $R_{\pi/2}$ occur: (1) in the NW corner ($R_\pi << R_{\pi/2}$) and (2) in the central area ($R_\pi >>$
$R_{\pi/2}$). The first area corresponds to a vegetated folded and faulted zone (Fig 3b-e).
Consequently, we consider the dataset poorly reliable in this location. In the central portion of
the study area, the difference between $R_\pi$ and $R_{\pi/2}$ (which increases as data dispersion
increases) is less pronounced in Mesh 1 than Mesh 2. Thus, data dispersion increases with
increasing the search window size (1200m for Mesh 1 and 1900m for Mesh 2), evidencing
that joint orientation is changing in this area. In the rest of the analyzed foreland, $R_{\pi/2}$ has
values similar to $R_\pi$, indicating approximately unimodal data distribution within this region,
and poor spatial organization of the longitudinal cross-joints.
Distribution of $Mv_\pi$ relates to the prevalence of NS-striking joints in the northern and
central portion of the study area. Towards the south, patches characterized by both N-S and
NW-SE-striking joints occur. High values of $R_\pi$ and $R_{\pi/2}$ are characteristic of such subareas,
which as previously mentioned, is indicative of unimodal joint trace distributions.
**Discussion**
Remotely sensed and mapped joint traces in the eastern portion of the Pyrenees-Ebro
system show systematic distributions of azimuthal orientations. In the frontal portion of the
belt and within the foredeep, joints are mostly transverse (i.e. N-S-striking), with limited
occurrence of E-W-striking longitudinal cross-joints. Approaching the southern border of the
foredeep, joints exhibit both N-S and NW-SE orientations where the pre-Cenozoic floor of
the foredeep is exposed. Joints affect the Lutetian to Priabonian foredeep infill, and are found
tilted together with strata within the Bellmunt anticline (Fig. 2b), which is Priabonian in age
(Burbank et al., 1992). Given the above, jointing in the study area must have taken place
during or before the Priabonian. The occurrence of the N-S striking joints within the Ebro
foredeep is documented also to the west of the study area (e.g. Turner and Hancock, 1990),
where joint emplacement affects up to the Miocene (Arlegui and Simon, 2001). Transverse
joints, striking approximately NNE-SSW occur also to the SW in Bartonian to Priabonian
strata cropping out at the boundary between the Ebro basin and the Catalan Coastal Ranges
(Alsaker et al., 1996). These data indicate that transverse joints systematically developed in
the foredeep basin of the E-W oriented Pyrenean belt. Locally, the process of transverse
jointing occurred until the early Miocene (i.e. until the end of mountain building). Also, pre-
thrusting transverse extensional faults of upper Paleocene to lower Eocene age occur a few

tens of km to the NE of the study area (Carrillo et al., 2020), being presently incorporated into the Pyrenean belt. Thus, we conclude that foredeep-parallel tension has established in the foredeep of the Pyrenean belt since the Paleocene and until the end of convergence. Transverse joints documented in this work clearly represent foredeep-related structures, which can develop by (i) N-S oriented layer-parallel shortening (LPS) or (ii) E-W oriented along foredeep stretching (Tavani et al., 2015). LPS is to be excluded, as the state of stress in this case would include a positive minimum stress (Fig. 1). In agreement, LPS-related extensional structures can form only due to fluid pressure contribution and they include mm- to cm-long fractures filled with calcite (which is removed from pressure solution seams, Fig. 1; Tavani et al., 2015 and references therein). The type (joints with no calcite infill) and size (tens of m-long) of transverse extensional structures described here are incompatible with the layer-parallel shortening mechanism.

The relatively constant orientation of joints along the strike of the foredeep, the occurrence of appreciable dispersion at the outer border of the foredeep, and the remarkably poor abundance of longitudinal joints allow us to derive two major conclusions:

(1) The almost linear trend of the eastern Pyrenees facilitates the exclusion of planar arching (e.g. Doglioni, 1995; Zhao and Jacobi, 1997) as the causative process for generating foredeep-parallel stretching (i.e. required to establish the negative $\sigma 3$ responsible for transverse jointing). Arching along the vertical plane parallel to the trench (Quintà and Tavani, 2012) represents instead a viable mechanism for generating along-foredeep stretching. This is analogous, albeit at a larger scale to the process of release faulting described by Destro (1995). The basic concept behind this mechanism is the following: when a straight line joining two fixed points - the tips of a fault in the case of Destro (1995) or the edges of the foredeep in the case of Quintà and Tavani (2012) – becomes an arch, there is stretching (Fig. 1b), which causes extensional stress parallel to the direction of elongation. In essence, this mechanism is expected to operate in any doubly plunging foredeep, particularly at its lateral edges, such as in the study area (Fig. 3a).

(2) Extension in the peripheral bulge, which is documented from many active and fossil foredeep basins (e.g. Bradley and Kidd, 1991; Ranero et al., 2003; Tavani et al., 2015; Granado et al., 2016b), including the lower Eocene foredeep basin presently incorporated into the Pyrenees (e.g. Martinez et al., 1989; Pujadas et al., 1989) appears to be weakly influential at the southern border of the study area (i.e. the upper Eocene peripheral bulge). Indeed, the

observed longitudinal joints are characteristically subordered, forming locally as cross-joints to the identified transverse set within the study area (Fig. 2d). The transverse set becomes less organized at the southern margin of the foredeep, where patches of N-S and NW-SE dominated domains do occur. This evidences the absence of a major forebulge-perpendicular extension capable of systematically reorienting σ3 at the external foredeep edge.

**Conclusions**

Analysis of remotely sensed and mapped joints in the eastern Pyrenees and in the adjacent Ebro foreland basin indicates that the emplacement of the dominant joint set within the area, which strikes perpendicular to the trend of the foredeep occurred prior to folding and developed in response to along-strike stretching caused by the plunging shape of the foredeep. Joints developed in response to flexuring of the lithosphere at the peripheral bulge do not occur in the area, suggesting that this mechanism has limited relevance to the observed joint system. This is confirmed by the variability of joint orientations observed at the foredeep external edge, negating the occurrence of a major forebulge-perpendicular extension able to systematically orient the stress field at the foredeep edge.

**Data availability**

Digitized traces in shapefile format and bedding and joint data in csv format are in the supplementary materials

**Author contributions**

ST, PG, AC, TS, JMC, and JAM contributed equally to the elaboration of the manuscript.

**Competing interests**

The authors declare that they have no conflict of interest.

**Acknowledgements**

We thank Roger Soliva, Eric Salomon, and an anonymous referee for useful suggestions. The Institut de Recerca Geomodels and the Geodinàmica i Anàlisi de Conques research group (2014SGR467SGR) acknowledges financial support from the Agència de Gestió d'Ajuts Universitaris i de Recerca (AGAUR) and the Secretaria d'Universitats i Recerca del Departament d'Economia i Coneixement de la Generalitat de Catalunya. Financial support form projects CGL2017-87631-P and PGC2018-093903-B-C22 from Ministerio de Ciencia, Innovación y Universidades/Agencia Estatal de Investigación/Fondo Europeo de Desarrollo Regional, Unión Europea is also acknowledged.

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

**Captions**

Figure 1

Scheme showing the architecture of a foreland fold-and-thrust belt and adjacent foredeep basin, with syn-orogenic fracture patterns in the different structural domains of the orogenic system. (A) The foredeep state of stress is governed by the permutation between the state of stress in the layer-parallel shortening and peripheral bulge domains. (B) The foredeep state of stress is controlled by the along-strike stretching of the foredeep.

Figure 2

Examples of pre-folding joints within the studied area. (A) N-S striking joint with plumose structures in the foredeep sediments (42°02'39.7"N; 2°13'54.9"E). (B) Tilted N-S and E-W striking joints in the southern limb of the Bellmunt anticline, with the red arrow indicating an E-W striking joint abutting on a N-S striking joint (42°05'39"N; 2°17'41.5"E). The density contours of poles to bedding and joints (in their present day orientation and after unfolding) refer to data collected in the Bellmunt anticline area. (C to F) Examples of joints seen on orthophotos. (C) Transverse joints. (D) N-S striking transverse joints with subordinate E-W striking cross-orthogonal joints. (E) NW-SE striking joints. (F) Rare example of multiple oblique sets occurring at the same exposure. Orthophotos are available from the Spanish Instituto Geográfico Nacional (https://pnoa.ign.es/).

Figure 3

(A) Simplified geological map of the western Pyrenees and Catalan Coastal Ranges (based on the Geological Map of Catalunya scale 1:250'000; https://www.icgc.cat/en/Downloads), with N-S geological cross-section (modified from Parés et al., 1999). (B) Detailed geological map of the study area, with digitized joints (C). (D) Orthophoto (https://pnoa.ign.es/) of the study area, with digitized joints (E). (F) Frequency distribution of joint traces trend and length.

Figure 4

Examples of joint patterns and resultant Mv and R parameters calculated for the $\pi$ and $\pi/2$ periods. For the five examples, we show the map view of the joints, the azimuthal frequency, and the sin-cos coordinates of the resultant values of Mv and R. Note that the distance from the center is proportional to R.

Figure 5
Results of circular statistics analysis for both Meshes 1 and 2. Length of traces of $Mv_\pi$ is
proportional to $R_\pi$. Color code refers to $R_\pi$ and $R_{\pi/2}$, whereas the orientation of traces is the
Mv. See text for details.

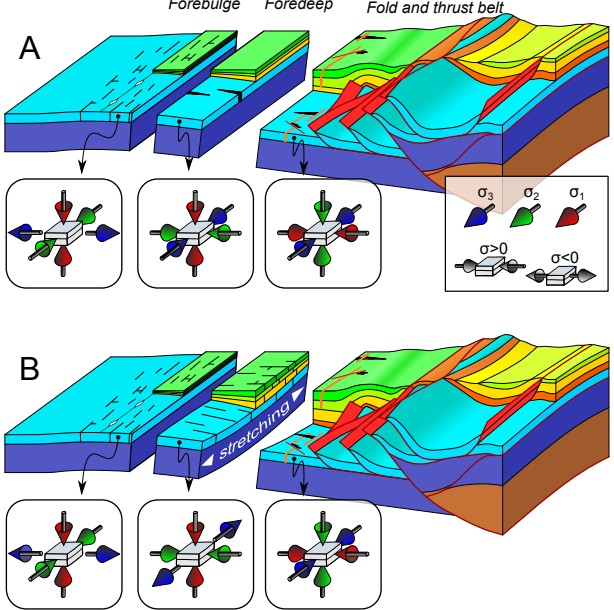

**Figure 1 (Single column)**
Scheme showing the architecture of a foreland fold-and-thrust belt and adjacent foredeep basin, with syn-orogenic fracture patterns in the different structural domains of the orogenic system. (A) The foredeep state of stress is governed by the permutation between the state of stress in the layer-parallel shortening and peripheral bulge domains. (B) The foredeep state of stress is controlled by the along-strike stretching of the foredeep.

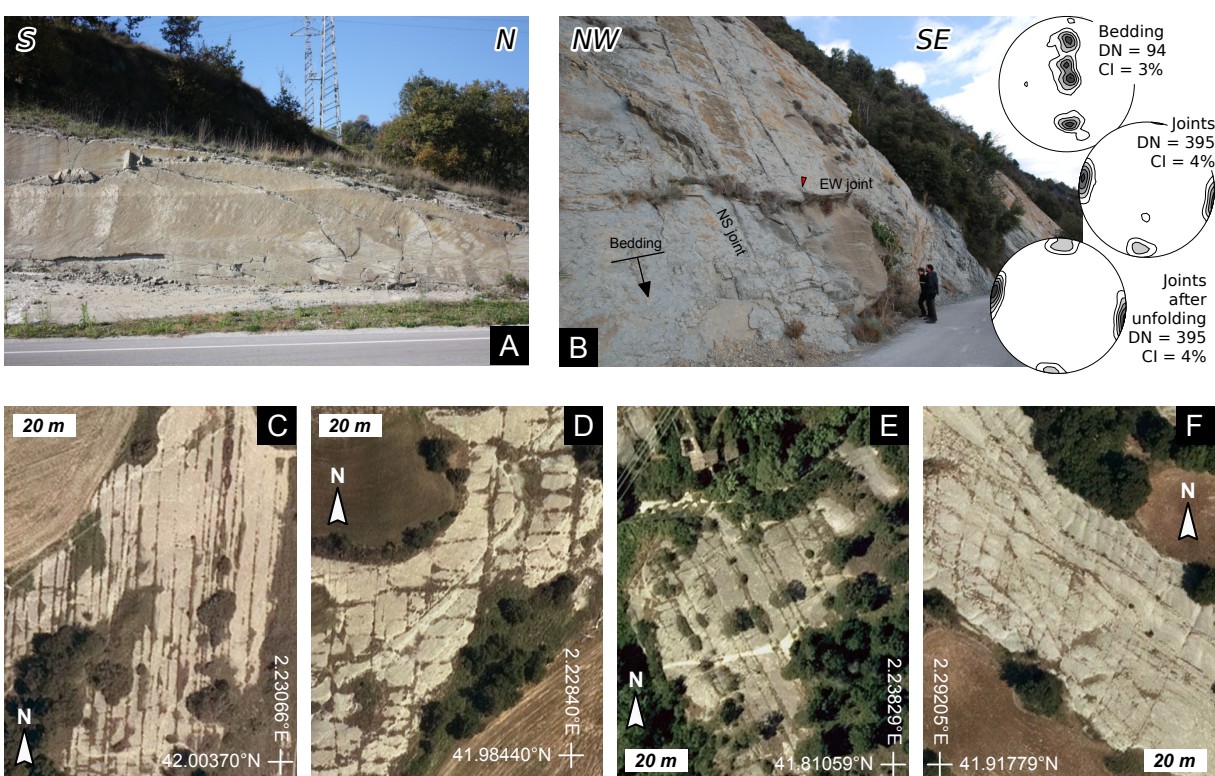

**Figure 2 (Double column)**

Examples of pre-folding joints within the studied area. (A) N-S striking joint with plumose structures in the foredeep sediments (42°02'39.7"N; 2°13'54.9"E). (B) Tilted N-S and E-W striking joints in the southern limb of the Bellmunt anticline, with the red arrow indicating an E-W striking joint abutting on a N-S striking joint (42°05'39"N; 2°17'41.5"E). The density contour of poles to bedding and joints (in their present day orientation and after unfolding) refer to data collected in the Bellmunt anticline area. (C to F) Examples of joints seen on orthophotos. (C) Transverse joints. (D) N-S striking transverse joints with subordinate E-W striking cross-orthogonal joints. (E) NW-SE striking joints. (F) Rare example of multiple oblique sets occurring at the same exposure. Orthophotos are available from the Spanish Instituto Geográfico Nacional (https://pnoa.ign.es/)..

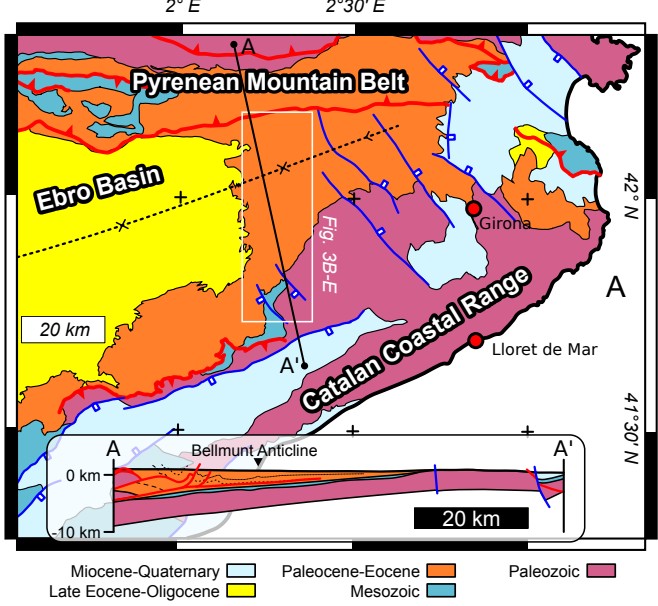

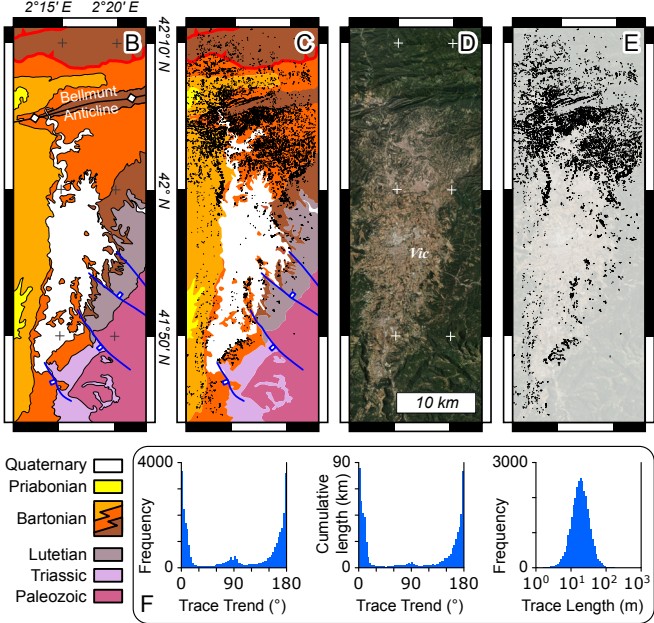

**Figure 3 (Single column)**
(A) Simplified geological map of the western Pyrenees and Catalan Coastal Ranges (based on the Geological Map of Catalunya scale 1:250'000; https://www.icgc.cat/en/Downloads), with N-S geological cross-section (modified from Parés et al., 1999). (B) Detailed geological map of the study area, with digitized joints (C). (D) Orthophoto (https://pnoa.ign.es/) of the study area, with digitized joints (E). (F) Frequency distribution of joint traces trend and length.

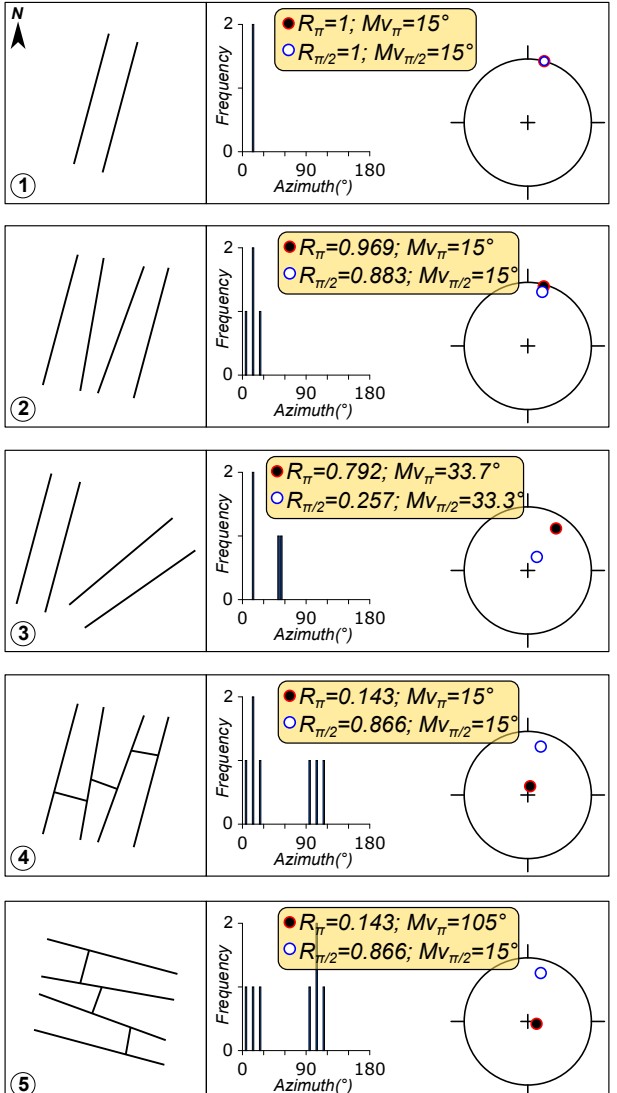

**Figure 4 (Single column)**
Examples of joint patterns and resultant Mv and R parameters calculated for the π and π/2 periods. For the five examples, we show the map view of the joints, the azimuthal frequency, and the sin-cos coordinates of the resultant values of Mv and R. Note that the distance from the center is proportional to R.

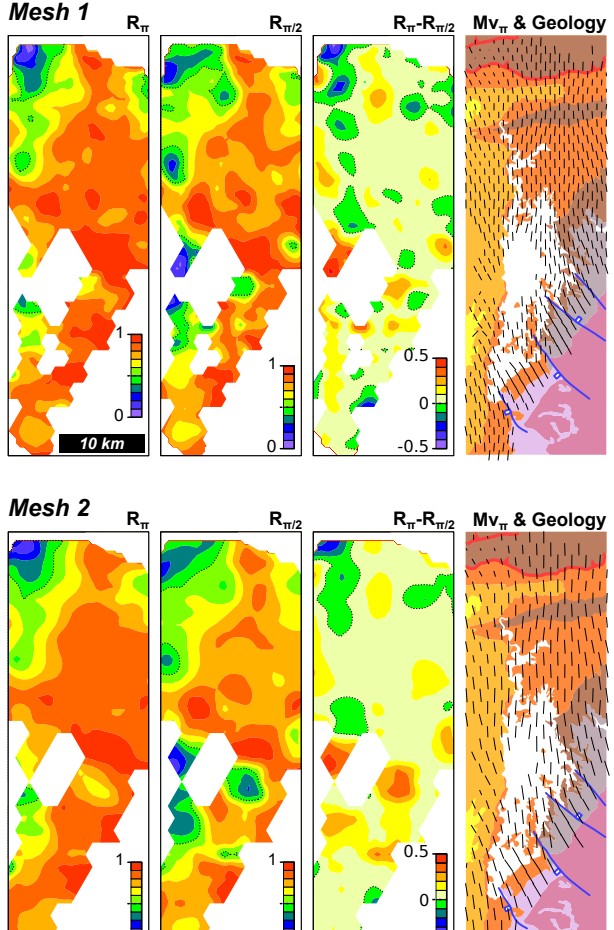

**Figure 5 (Single column)**

Results of circular statistics analysis for both Meshes 1 and 2. Length of traces of Mvπ is proportional to Rπ. Color code refers to Rπ and Rπ/2, whereas the orientation of traces is the Mv. See text for details. .