# Peer review of "Transverse jointing in foreland fold-and-thrust belts: a remote sensing"

_Solid Earth, 2020_

## Referee Comment (RC1) · Anonymous Referee #1 · 2 Jun 2020

The manuscript entitled "Transverse jointing in foreland fold-and-thrust belts: a remote sensing analysis in the eastern Pyrenees" reports high quality data derived from orthophotos analysis of joints in the Pyrenean Ebro Basin. Joint pattern is then presented in an unparalleled way as it covers basin-scale width, and the role of the foredeep-forebulge onto the upper crust is discussed. Data are clearly presented, some minor information are missing, and the figures are overall of high quality. Putting aside the editorial choice of including this study in a special issue about fluids, fractures and faults while fluids are out of the scope of this manuscript, it still suffer some important points that needs to be adressed. 1) The first one is that the overall interpretation seems heavily model driven. Indeed the E-W fractures are interpreted as forebulge-parallel

extension, which make sense, but the systematic attribution of the N-S fractures to accros strike extension can be argued against: - an alternate interpretation would be to consider the N-S fractures as related to LPS, postponing the E-W, forebulge related fractures, leading to similar patterns than the one described. The occurence of a NNE-SSW (what is the mean strike of it?) goes well into this alternate scenario, as the Ebro Basin underwent a regional 20° Clockwise rotation during paleogene, as reconstructed by the paleomagnetic data (Parès et al., 1988, Physics of the Earth and Planetary Interiors, Volume 52, Issue 3-4, p. 267-282). This rotation does not seem to have been considered by the authors, and I think this needs adressed. Two important things are missing to back up the interpretation of the authors: relative chronology; and observation and report of systematic occurence of N-S joints with E-W joints. 2) I would be interested to see reported the length of the fracture tracks for each set, I am sure it could be of interest as well to solve the problem I mentionned in my first comment.

There is also minor remarks: Page 2, line 26-27:"Even in arched systems, the forebulge, the foredeep, and the belt tend to be nearly parallel to each other locally" –> can you report related references?

Page 4, line 28-29: "The NE and SE portions of the study area are highly vegetated (Fig. 3d,e) and only a few joint traces have been mapped there." –> how does it affect the statistic? Why not leaving these out?

Page 5: Why did you choose these lenghts for the triangular mesh? Do you need it to be one order of magnitude longer than the longest fractures? Can you discuss the impact?

Figure 2 C-F: The north is not really clear from this representation.

---

## Author Comment (AC1) · 5 Jun 2020

**Comment**

it [the ms.] still suffer some important points that needs to be adressed. 1) The first one is that the overall interpretation seems heavily model driven. Indeed the E-W fractures are interpreted as forebulge-parallel extension, which make sense, but the systematic attribution of the N-S fractures to accros strike extension can be argued against: - an alternate interpretation would be to consider the N-S fractures as related to LPS, postponing the E-W, forebulge related fractures, leading to similar patterns than the one described.

[Figure]

**Response**
We thank the reviewer for this comment. We will add this text in the revised version "These could be also interpreted as layer parallel shortening (LPS)-related transverse extensional structures. However, LPS-related extensional structures include mm- to cm-long fractures filled with calcite (which is removed from pressure solution seams; Tavani et al., 2015 and references therein). The type (joints with no calcite infill) and size (tens of m-long) of transverse extensional structures described here are incompatible with layer-parallel shortening.

**Comment**
The occurence of a NNESSW (what is the mean strike of it?) goes well into this alternate scenario, as the Ebro Basin underwent a regional 20âŬȩ Clockwise rotation during paleogene, as reconstructed by the paleomagnetic data (Parès et al., 1988, Physics of the Earth and Planetary Interiors, Volume 52, Issue 3-4, p. 267-282). This rotation does not seem to have been considered by the authors, and I think this needs adressed.

**Response**
It has been recently demonstrated (after the initial papers describing paleomagnetic data in the Triassic red beds) that the Ebro basin has not experienced a general rotation during the Paleogene. Paleogene vertical-axis rotations in the Pyrenees are mainly related with displacement gradients of the thrust sheets, mostly resulting from the distribution of the Triassic salt detachment horizon (Sussman et al., 20014; Soto et al., 2006; Muñoz et al., 2013). In addition, older vertical axis rotations, can be related with the extensional and sinistral displacement of Iberia during Early Cretaceous (Dinarès-Turell and García-Senz, 2000; Gong et al., 2009). Apart from these vertical axis rotations, which are at present well documented, the Ebro basin and in detailed the eastern part of the Ebro basin where this is study has been located has not experienced any vertical-axis rotation as documented by paleomagnetic studies (Burbank et al., 1992; Taberner et al., 1999).

Burbank, D. W., Puigdefàbregas, C., Muñoz, J. A. (1992). The chronology of the Eocene tectonic and stratigraphic development of the eastern Pyrenean foreland basin, northeast Spain. Geological Society of America Bulletin, 104(9), 1101–1120. http://doi.org/10.1130/0016-7606(1992)104<1101:TCOTET>2.3.CO;2

Dinarès-Turell, J., Senz, J. G. (2000). Remagnetization of Lower Cretaceous limestones from the southern Pyrenees and relation to the Iberian plate geodynamic evolution. Journal of Geophysical Research: Solid Earth (1978–2012), 105(B8), 19405–19418. http://doi.org/10.1029/2000JB900136

Gong, Z., van Hinsbergen, D. J. J., Vissers, R. L. M., Dekkers, M. J. (2009). Early Cretaceous syn-rotational extension in the Organyà basin—New constraints on the palinspastic position of Iberia during its rotation. Tectonophysics, 473(3-4), 312–323. http://doi.org/10.1016/j.tecto.2009.03.003

Soto, R., Casas-Sainz, A. M., Pueyo, E. L. (2006). Along-strike variation of orogenic wedges associated with vertical axis rotations. Journal of Geophysical Research: Solid Earth, 111(B10), B10402. http://doi.org/10.1029/2005JB004201

Sussman, A. J., Butler, R. F., Dinarès-Turell, J., Vergés, J. (2004). Vertical-axis rotation of a foreland fold and implications for orogenic curvature: an example from the Southern Pyrenees, Spain. Earth and Planetary Science Letters, 218(3-4), 435–449. http://doi.org/10.1016/S0012-821X(03)00644-7

Taberner, C., Dinarès-Turell, J., Giménez, J., Docherty, C. (1999). Basin infill architecture and evolution from magnetostratigraphic cross-basin correlations in the southeastern Pyrenean foreland basin. Geological Society of America Bulletin, 111(8), 1155.

**Comment**

Two important things are missing to back up the interpretation of the authors: relative

chronology; and observation and report of systematic occurence of N-S joints with E-W joints.

**Response**

The few E-W striking joints systematically abut on the N-S striking set, indicating that E-W striking joints are cross-joints formed perpendicular to the master (N-S) joint set. This is well shown in figures 2D and 2E (for the NNW-SSE striking set), and it will be mentioned in the revised version.

**Comment**

I would be interested to see reported the length of the fracture tracks for each set, I am sure it could be of interest as well to solve the problem I mentionned in my first comment.

**Response**

This graph will be added

There is also minor remarks:

**Comment**

Page 2, line 26-27:"Even in arched systems, the forebulge, the foredeep, and the belt tend to be nearly parallel to each other locally" –> can you report related references?

**Response**

References will be added

**Comment**

Page 4, line 28-29: "The NE and SE portions of the study area are highly vegetated (Fig. 3d,e) and only a few joint traces have been mapped there." –> how does it affect

the statistic? Why not leaving these out?

**Response**

We agree. Nodes with < 20 data have not been considered in our analysis. This will be mentioned in the new version of the ms.

**Comment**

Page 5: Why did you choose these lenghts for the triangular mesh? Do you need it to be one order of magnitude longer than the longest fractures? Can you discuss the impact?

**Response**

The radius of the circular moving window is set to these values for these two reasons: 1) it is two orders of magnitude longer than the average length of joints; 2) it is larger enough to ensure that at each node the data number is >20

**Comment**

Figure 2 C-F: The north is not really clear from this representation.

**Response**

we will add the north.

---

## Referee Comment (RC2) · Eric Salomon (Referee) · 9 Jun 2020

This manuscript deals with the character of jointing in the foreland of the Pyrenees through mapping their trend in aerial imagery. The manuscript describes a pronounced N-S trend of joints, i.e. perpendicular to the trend of the Pyrenees and provides an explanation for their formation. I navigated myself across this area with Google Earth and it is really fascinating to see the clarity and spatial extend of this joint system making also this manuscript, which is very well written and accompanied by nice figures, an interesting read.

Nevertheless, before accepting the manuscript for publication, I think a few improvements or clarifications need to be made.

Please allow me to be a bit provocative at the beginning of this review. In the introduction, you outline a mechanism to explain transverse jointing in the foreland. Then you continue with the outlook that the acquired remote sensing data allows to investigate the primary mechanism for this joint formation. Hence I wonder: Haven't you proposed already before the data collection and discussion of what the primary mechanism is? Does this lead to a bias in the data interpretation? Maybe it would be better to present the model after the data presentation in the discussion to avoid that such an impression might arise.

Also on that regard, a discussion on the potential driving force behind the suggested orogen-parallel stretching of the foreland basin is largely missing (or well hidden, in case I missed it), which would be very interesting though. In the introduction, it is briefly referred to two publications invoking the possibility of lithospheric bending to account for such kind of stretching (Doglioni, 1995; Quintà and Tavani, 2012; although the process described by Doglioni is regarded as not applicable for the study area, which is comprehensible). In the discussion, this subject is covered with only two sentences (page 7 line 31 to page 8 line 3; half of it being a repetition of the introduction statement). Here, foreland-parallel stretching is suggested to form the N-S joints and an analogue reference is made to the process of release faulting (Destro, 1995). I think this requires much more attention: Destro (1995) describes a purely extensional setting and it is therefore not straightforward to understand how this applies to the Pyrenean foreland, especially in the light that you propose foreland-parallel extension from the Paleocene until the end of convergence (page 7, line 22). Hence, I believe a more elaborate discussion for the use of this model is necessary, in particular, and potential driving mechanism for such stretching, in general.

Adding to this, you mention the westward plunge of the foredeep basin and refer to figure 3a. I am not sure if this is it actually visible in the figure or if it requires previous knowledge of the region to identify it!? I think an E-W cross-section would be very helpful.

A second issue revolves around the timing of joint formation. You state that the dominant N-S trending joint system formed prior to folding and refer to figure 2b, where joints are supposedly tilted. Unfortunately, from the picture alone, it is very hard to see this. How did you determine that these joints are tilted? How can you exclude the possibility of joint formation after folding? Such a determination appears to me as a very difficult asset, since you would have to know their original orientation and at the same time line out why its present orientation is not the original one. I think this is a very important issue that needs to be clarified.

A second argument for the age of joints is their absence in Quaternary sediments. First, there is still a large age span from the Quaternary to the Eocene (using the word "evidence" (page 7, line 9) for an Eocene formation age is therefore maybe a stretch), and second: what is the character of these sediments? Are they solidified to a degree where fractures would be able to form in case the joints in the Eocene rock were of Quaternary age?

Another thing: As you have been in the field, it would be great to see a comparison of field data with the remote lineament data. E.g. do the joints have a preferential dip direction, are they all just vertical?

Some other minor comments to the figures:
Figure 1: This is a very nice figure, but some features can only be identified when

zooming in a lot, i.e. the text "peripheral bulge", veins, and stylolites. Please improve this. Also, I recommend to place the names forebulge, foredeep, foreland fold-and-thrust belt into/above the block figure and not just mention them in the figure caption.

Figure 2: please show the locations of these outcrops in figure 3. Also, I would prefer to show field photos after showing a map of the study area. In Figure 2b, please point at the joints as it is not super clear that the big surface is, I assume, the bedding surface.

Figure 3: add some placemarks (e.g. towns) to the map, so that it's a bit easier for the reader to capture the location of the study area. (took me a little bit to find the exact area on google earth).

Figure 5: I think it would be really nice, if you exemplarily show a few rose plots (joint length-weighted) for different colored regions in figure 5. I believe this would make it much easier for the reader to understand how to read the color code of the figure.

I hope these comments are useful for you.
Eric Salomon

---

## Editor Comment (EC1) · Roger Soliva (Editor) · 9 Jun 2020

Dear Authors,

We now have two reviews of your manuscript suggesting moderate revision. I encourage you to also answer point per point to the second reviewer and carefully look at the way you present the interpretation of the data. Both reviewers mentioned that the approach is too much model driven and should consider and discuss any possible process, and the relevant litterature. Thanks in advance for the efforts made and the next revisions.

Best regards, Roger Soliva

---

## Editor Comment (EC2) · Roger Soliva (Editor) · 26 Jun 2020

Dear Authors,

Thank you for the revisions made which improve the quality of the paper. I understand the wish to keep hypothesis first in the paper (model based writing style mentionned by both reviewers and I) and then I have recommendations about strengthening both your hypothesis and interpretation. This is needed anyway in the paper wathever the syle is chosen, but actually more relevant if you chose this style. Especially, I recommend considering some significant references on the topic that have been ignored to reinforce both your interpretations and hypotheses.

[Figure]

1) In the revised version, you mention in lines 30-31 of p7 "we conclude that foredeep-parallel extension has occurred in the foredeep of the Pyrenean belt since the Pale-ocene and until the end of convergence" Do you consider here that Sigma 3 is negative as proposed in Figure 1b and introduction ? Extension is an unclear deformation term not synonym to tension or extensional stresses (i.e. negative stresses). Clarify this in the text please.

2) On this negative stresses as shown in Figure 1b, although we can agree on your interpretation, the paper suffers considering the significant contributions from experimental tests which have been compared to natural joints from the past decade. You mention extensional stresses (negative) but what about splitting without negative stresses (and even with a slightly compressive sigma 3) such as demonstrated in dry axi-symmetric, oedometric, plane strain and poly-axial experiments by Chemenda et al. (JGR,2011) and Jorand et al (Tectonophysics 2012) ? These studies shows joints formed under dry contraction without negative sigma 3, which are not so far than uniaxial splitting fractures observed in triaxial cells (e.g. Holzhausen and Johnson, 1979), but here clearly without the triaxial boundary effect mentionned by Fakhimi and Hemami (2015).

3) A common species of joints show very low displacement gradients compared to other fractures (veins, faults) (Pollard and Aydin, 1998; Schultz et al., 2008), which also support the general fact that joint sets do not require significant amount of negative stresses perpendicular to them. Have you measured the mean opening of the observed fractures ? Can this help to discuss this point ?

4) I recommend you to better support the hypothesis mentionned in lines 31-32 p2 and 1-3 p3, which only relies on one reference, while others works previously described stress permutation during LPS. For example, stress permutation in foreland basin has been proposed from field observations and stress path calulations by Soliva et al. (2013), and reused with nearly the same concept in Fossen's book 2015 version. Addition of such references is just a fair strengthening of the hypothesis on which the work relies.

Please, consider discussing/including these elements in your paper, you have plenty of space since the paper is quite short.

Best regards, Roger Soliva

---

## Author Comment (AC3) · 30 Jun 2020

Comment 1
In the revised version, you mention in lines 30-31 of p7 "we conclude that foredeep parallel extension has occurred in the foredeep of the Pyrenean belt since the Paleocene and until the end of convergence" Do you consider here that Sigma 3 is negative as proposed in Figure 1b and introduction ? Extension is an unclear deformation term not synonym to tension or extensional stresses (i.e. negative stresses). Clarify this in the text please.

[Figure]

Response
We consider sigma 3 negative. The new text is: "Thus, we conclude that foredeep-parallel tension has established in the foredeep of the Pyrenean belt since the Paleocene and until the end of convergence"

Comment 2
On this negative stresses as shown in Figure 1b, although we can agree on your interpretation, the paper suffers considering the significant contributions from experimental tests which have been compared to natural joints from the past decade. You mention extensional stresses (negative) but what about splitting without negative stresses (and even with a slightly compressive sigma 3) such as demonstrated in dry axi-symmetric, oedometric, plane strain and poly-axial experiments by Chemenda et al. (JGR,2011) and Jorand et al (Tectonophysics 2012) ? These studies shows joints formed under dry contraction without negative sigma 3, which are not so far than uniaxial splitting fractures observed in triaxial cells (e.g. Holzhausen and Johnson, 1979), but here clearly without the triaxial boundary effect mentionned by Fakhimi and Hemami (2015).

Response
We have no doubt that it is possible to replicate the morphology of a joint at the specimen size in an experimental apparatus using a Sigma3 > 0. However, we have some concerns about the possibility of upscaling such an experimental result at the basin scale and for tens of meters long systematic joints. Also, the occurrence of orthogonal cross-joints is not compatible with compressive sigma 3. We have added this text: "This indicates that E-W striking joints are cross-joints formed perpendicular to, and about synchronously with, the N-S striking joint set and that N-S joints formed in response to a negative (tensile) minimum stress (e.g. Bai and Gross, 1999; Bai et al., 2002)"

Comment 3

A common species of joints show very low displacement gradients compared to other fractures (veins, faults) (Pollard and Aydin, 1998; Schultz et al., 2008), which also support the general fact that joint sets do not require significant amount of negative stresses perpendicular to them. Have you measured the mean opening of the observed fractures ? Can this help to discuss this point ?

Response
We have not collected joint aperture data

Comment 4
I recommend you to better support the hypothesis mentionned in lines 31-32 p2 and 1-3 p3, which only relies on one reference, while others works previously described stress permutation during LPS. For example, stress permutation in foreland basin has been proposed from field observations and stress path calulations by Soliva et al. (2013), and reused with nearly the same concept in Fossen's book 2015 version. Addition of such references is just a fair strengthening of the hypothesis on which the work relies

Response
We have added this text " This is evidenced by the occurrence of bedding-perpendicular pressure solution-vein pairs (e.g Railsback and Andrews, 1995; Evans and Elmore, 2006; Quintà and Tavani, 2012; Weil and Yonkee, 2012) and/or conjugate strike-slip faults at a high angle to bedding (e.g. Marshak et al., 1982, Hancock, 1985, Erslev, 2001; Lacombe et al., 2006; Amrouch et al., 2010, Weil and Yonkee, 2012) occurring in foreland areas and in the adjacent fold and thrust belts worldwide, although in many cases structures associated with this strike-slip regime do not develop during layer parallel shortening (Soliva et al., 2013). "

---

## Author Response (AR1)

**RESPONSE TO REVIEWERS**

**Reviewer 1**

**Comment 1**

it [the ms.] still suffer some important points that needs to be adressed. 1) The first one is that the overall interpretation seems heavily model driven. Indeed the E-W fractures are interpreted as forebulge-parallel extension, which make sense, but the systematic attribution of the N-S fractures to accros strike extension can be argued against: - an alternate interpretation would be to consider the N-S fractures as related to LPS, postponing the E-W, forebulge related fractures, leading to similar patterns than the one described.

**Response: Done.**

*We thank the reviewer for this comment. We have added this text:*

*"LPS is to be excluded, as the state of stress in this case would include a positive minimum stress (Fig. 1). In agreement, LPS-related extensional structures can form only due to fluid pressure contribution and they include mm- to cm-long fractures filled with calcite (which is removed from pressure solution seams, Fig. 1; Tavani et al., 2015 and references therein). The type (joints with no calcite infill) and size (tens of m-long) of transverse extensional structures described here are incompatible with the layer-parallel shortening mechanism.*

**Comment 2**

The occurence of a NNESSW (what is the mean strike of it?) goes well into this alternate scenario, as the Ebro Basin underwent a regional 20◦ Clockwise rotation during paleogene, as reconstructed by the paleomagnetic data (Parès et al., 1988, Physics of the Earth and Planetary Interiors, Volume 52, Issue 3-4, p. 267-282). This rotation does not seem to have been considered by the authors, and I think this needs adressed.

**Response: Not agreed.**

*It has been recently demonstrated (after the initial papers describing paleomagnetic data in the Triassic red beds) that the Ebro basin has not experienced a general rotation during the Paleogene. Paleogene vertical-axis rotations in the Pyrenees are mainly related with displacement gradients of the thrust sheets, mostly resulting from the distribution of the Triassic salt detachment horizon (Sussman et al., 20014; Soto et al., 2006; Muñoz et al., 2013). In addition, older vertical axis rotations, can be related with the extensional and sinistral displacement of Iberia during Early Cretaceous (Dinarès-Turell and García-Senz, 2000; Gong et al., 2009).*

*Apart from these vertical axis rotations, which are at present well documented, the Ebro basin and in detailed the eastern part of the Ebro basin where this is study has been located has not experienced any vertical-axis rotation as documented by paleomagnetic studies (Burbank et al., 1992; Taberner et al., 1999).*

*Burbank, D. W., Puigdefàbregas, C., & Muñoz, J. A. (1992). The chronology of the Eocene tectonic and stratigraphic development of the eastern Pyrenean foreland basin, northeast Spain. Geological Society of America Bulletin, 104(9), 1101–1120. http://doi.org/10.1130/0016-7606(1992)104<1101:TCOTET>2.3.CO;2*

*Dinarès-Turell, J., & Senz, J. G. (2000). Remagnetization of Lower Cretaceous limestones from the southern Pyrenees and relation to the Iberian plate geodynamic evolution. Journal of Geophysical Research: Solid Earth (1978–2012), 105(B8), 19405–19418. http://doi.org/10.1029/2000JB900136*

*Gong, Z., van Hinsbergen, D. J. J., Vissers, R. L. M., & Dekkers, M. J. (2009). Early Cretaceous syn-rotational extension in the Organyà basin—New constraints on the palinspastic position of Iberia during its rotation. Tectonophysics, 473(3-4), 312–323. http://doi.org/10.1016/j.tecto.2009.03.003*

*Soto, R., Casas-Sainz, A. M., & Pueyo, E. L. (2006). Along-strike variation of orogenic wedges associated with vertical axis rotations. Journal of Geophysical Research: Solid Earth, 111(B10), B10402. http://doi.org/10.1029/2005JB004201*

*Sussman, A. J., Butler, R. F., Dinarès-Turell, J., & Vergés, J. (2004). Vertical-axis rotation of a foreland fold and implications for orogenic curvature: an example from the Southern Pyrenees, Spain. Earth and Planetary Science Letters, 218(3-4), 435–449. http://doi.org/10.1016/S0012-821X(03)00644-7*

*Taberner, C., Dinarès-Turell, J., Giménez, J., & Docherty, C. (1999). Basin infill architecture and evolution from magnetostratigraphic cross-basin correlations in the southeastern Pyrenean foreland basin. Geological Society of America Bulletin, 111(8), 1155.*

**Comment 3**
Two important things are missing to back up the interpretation of the authors: relative chronology; and observation and report of systematic occurence of N-S joints with E-W joints.
**Response: Done**
*The few E-W striking joints systematically abut on the N-S striking set, indicating that E-W striking joints are cross-joints formed perpendicular to the master (N-S) joint set. This is well shown in figures 2D and 2E (for the NNW-SSE striking set), and it is now mentioned in the revised text.*

**Comment 4**
I would be interested to see reported the length of the fracture tracks for each set, I am sure it could be of interest as well to solve the problem I mentionned in my first comment.
**Response: Done**
*Graph added in figure 3f*

There is also minor remarks:
**Comment 5**
Page 2, line 26-27:"Even in arched systems, the forebulge, the foredeep, and the belt tend to be nearly parallel to each other locally" –> can you report related references?
**Response: Done**
*Added*

**Comment 6**
Page 4, line 28-29: "The NE and SE portions of the study area are highly vegetated (Fig. 3d,e) and only a few joint traces have been mapped there." –> how does it affect the statistic? Why not leaving these out?
**Response: Done**
*We agree. Nodes with < 20 data have not been considered in our analysis. This is now mentioned in the text.*

**Comment 7**
Page 5: Why did you choose these lenghts for the triangular mesh? Do you need it to be one order of magnitude longer than the longest fractures? Can you discuss the impact?
**Response: Done**
*The radius of the circular moving window is set to these values for these two reasons: 1) it is two orders of magnitude longer than the average length of joints; 2) it is larger enough to ensure that at each node the data number is >20. This is now explicated in the text*

**Comment 8**

Figure 2 C-F: The north is not really clear from this representation.
**Response: Done**
*North added.*

**Reviewer 2**

**Comment 1**
In the introduction, you outline a mechanism to explain transverse jointing in the foreland. Then you continue with the outlook that the acquired remote sensing data allows to investigate the primary mechanism for this joint formation. Hence I wonder: Haven't you proposed already before the data collection and discussion of what the primary mechanism is? Does this lead to a bias in the data interpretation? Maybe it would be better to present the model after the data presentation in the discussion to avoid that such an impression might arise.

**Response: Not agreed**.
This is a matter of writing style and not a source of bias. In our view the rationale of the work and the state of the art must include a brief introduction to the causative geological processes allowing to understand data.

**Comment 2**
Also on that regard, a discussion on the potential driving force behind the suggested orogen-parallel stretching of the foreland basin is largely missing (or well hidden, in case I missed it), which would be very interesting though.
**Response: Done**
We have added this text in the discussion: *The basic concept behind this mechanism is the following: when a straight line joining two fixed points - the tips of a fault in the case of Destro (1995) or the edges of the foredeep in the case of Quintà and Tavani (2012) – becomes an arch, there is stretching (Fig. 1b), which causes extensional stress parallel to the direction of elongation. In essence, this mechanism is expected to operate in any doubly plunging foredeep, particularly at its lateral edges, such as in the study area (Fig. 3a)*

**Comment 3**
In the introduction, it is briefly referred to two publications invoking the possibility of lithospheric bending to account for such kind of stretching (Doglioni, 1995; Quintà and Tavani, 2012; although the process described by Doglioni is regarded as not applicable for the study area, which is comprehensible). In the discussion, this subject is covered with only two sentences (page 7 line 31 to page 8 line 3; half of it being a repetition of the introduction statement). Here, foreland-parallel stretching is suggested to form the N-S joints and an analogue reference is made to the process of release faulting (Destro, 1995). I think this requires much more attention: Destro (1995) describes a purely extensional setting and it is therefore not straightforward to understand how this applies to the Pyrenean foreland, especially in the light that you propose foreland-parallel extension from the Paleocene until the end of convergence (page 7, line 22). Hence, I believe a more elaborate discussion for the use of this model is necessary, in particular, and potential driving mechanism for such stretching, in general.
**Response: Done**
See response to the previous point.

**Comment 4**
Adding to this, you mention the westward plunge of the foredeep basin and refer to figure 3a. I am not sure if this is it actually visible in the figure or if it requires previous knowledge of the region to identify it!? I think an E-W cross-section would be very helpful.
**Response: Done**

We have added the trace of the axis of the foredeep basin in figure 3 to show its W-ward plunge.

**Comment 5**
A second issue revolves around the timing of joint formation. You state that the dominant N-S trending joint system formed prior to folding and refer to figure 2b, where joints are supposedly tilted. Unfortunately, from the picture alone, it is very hard to see this. How did you determine that these joints are tilted? How can you exclude the possibility of joint formation after folding? Such a determination appears to me as a very difficult asset, since you would have to know their original orientation and at the same time line out why its present orientation is not the original one. I think this is a very important issue that needs to be clarified.
**Response: Done**
Timing of deformation is rather evident from figure 2. We have improved the description of this figure: *In the field, joints are constantly bedding-perpendicular, regardless of the bedding dip (Fig. 2a,b), and they are characterized by the occurrence of either a single set (Fig. 2c) or by a ladder pattern (Fig. 2d,e). In the latter case, the few E-W striking joints are almost everywhere perpendicular to the N-S striking set and abut on it (Fig. 2e). This indicates that E-W striking joints are cross-joints formed perpendicular to, and about synchronously with, the N-S striking joint set*

**Comment 6**
A second argument for the age of joints is their absence in Quaternary sediments. First, there is still a large age span from the Quaternary to the Eocene (using the word "evidence" (page 7, line 9) for an Eocene formation age is therefore maybe a stretch), and second: what is the character of these sediments? Are they solidified to a degree where fractures would be able to form in case the joints in the Eocene rock were of Quaternary age?
**Response: Done**
See response to previous point.

**Comment 7**
Another thing: As you have been in the field, it would be great to see a comparison of field data with the remote lineament data. E.g. do the joints have a preferential dip direction, are they all just vertical?
**Response: Done**
We have added stereoplots of joints collected in the northern portion of the study area.

**Comment 8**
Figure 1: This is a very nice figure, but some features can only be identified when zooming in a lot, i.e. the text "peripheral bulge", veins, and stylolites. Please improve this. Also, I recommend to place the names forebulge, foredeep, foreland fold-andthrust belt into/above the block figure and not just mention them in the figure caption.
**Response: Done**
Labels added.

**Comment 9**
Figure 2:
1) please show the locations of these outcrops in figure 3.
2) Also, I would prefer to show field photos after showing a map of the study area.
3) In Figure 2b, please point at the joints as it is not super clear that the big surface is, I assume, the bedding surface.
**Response: Partly agreed**
1) This cannot be done due to the size of the figure: the labels of the five sites would cover much of the figure.
2) In the text the figure 2 is called before figure 3, so it cannot be shown before.

3) Yes, the south-dipping surface is the bedding, this is now mentioned.

**Comment 10**
Figure 3: add some placemarks (e.g. towns) to the map, so that it's a bit easier for the reader to capture the location of the study area. (took me a little bit to find the exact area on google earth).
**Response. Done**.
Added

**Comment 11**
Figure 5: I think it would be really nice, if you exemplarily show a few rose plots (joint length-weighted) for different colored regions in figure 5. I believe this would make it much easier for the reader to understand how to read the color code of the figure.
**Response: Done**
This is probably a misunderstanding. The colour code refers to the dispersion of azimuthal data, which is not well appreciable in rose diagrams as the dominant set is much developed than the other sets. We have improved the caption of the figure.

[revised manuscript text omitted]